# A LAT1-Like Amino Acid Transporter Regulates Neuronal Activity in the *Drosophila* Mushroom Bodies

**DOI:** 10.3390/cells13161340

**Published:** 2024-08-13

**Authors:** Julie Delescluse, Mégane M. Simonnet, Anna B. Ziegler, Kévin Piffaretti, Georges Alves, Yael Grosjean, Gérard Manière

**Affiliations:** 1Centre des Sciences du Goût et de l’Alimentation, CNRS, INRAe, Institut Agro, Université de Bourgogne, F-21000 Dijon, France; 2Institute for Neuro- and Behavioral Biology, University of Münster, 48149 Münster, Germany

**Keywords:** solute carrier, SLC, LAT-1, amino acids, mushroom bodies, *Drosophila*

## Abstract

The proper functioning of neural circuits that integrate sensory signals is essential for individual adaptation to an ever-changing environment. Many molecules can modulate neuronal activity, including neurotransmitters, receptors, and even amino acids. Here, we ask whether amino acid transporters expressed by neurons can influence neuronal activity. We found that *minidiscs* (*mnd*), which encodes a light chain of a heterodimeric amino acid transporter, is expressed in different cell types of the adult *Drosophila* brain: in mushroom body neurons (MBs) and in glial cells. Using live calcium imaging, we found that MND expressed in α/β MB neurons is essential for sensitivity to the L-amino acids: Leu, Ile, Asp, Glu, Lys, Thr, and Arg. We found that the Target Of Rapamycin (TOR) pathway but not the Glutamate Dehydrogenase (GDH) pathway is involved in the Leucine-dependent response of α/β MB neurons. This study strongly supports the key role of MND in regulating MB activity in response to amino acids.

## 1. Introduction

For any living organism, such as *Drosophila melanogaster*, an adequate supply of nutrients, especially amino acids (AAs), is essential to ensure a variety of functions including development [1], growth [2], lifespan and survival [3,4,5], reproduction [6,7], egg production and fecundity [3,4,8], and sleep [9]. While some AAs can be synthesized endogenously from precursors, others, known as essential amino acids (EAAs), such as Leu, Ile, Thr, are supplied by the diet. They also provide energy to fuel cellular metabolism. Some AAs are precursors of hormones, such as tryptophan and tyrosine, which are used to produce melatonin and noradrenaline/adrenaline, respectively [10,11]. Some AAs are also neurotransmitter precursors, such as tryptophan, which is required for serotonin production; tyrosine is required for L-dopa and dopamine production [12,13,14]. Others, such as glutamate, are used directly as neurotransmitters [15], and glutamate is also the precursor of the neurotransmitter GABA [16].

Mammals and invertebrates, such as *Drosophila*, are able to maintain an appropriate AA balance by selecting diets that contain the EAAs they need and avoiding diets that lack EAAs [17,18]. In *Drosophila* larvae, an unbalanced AA diet is detected by dopaminergic neurons (DANs) in the brain via an intracellular AA sensor: the serine/threonine kinase General Control Nonderepressible 2 (GCN2) which acts upstream of GABA signaling to inhibit it and promotes avoidance of an EAA-deficient diet [18]. In adults, food intake is also promoted by three AAs: Glu, Asp, and Ala [19]. EAAs such as Leu and Ile are required for the release of insulin-like peptides (DILPs) by insulin-producing cells (IPCs) in the brain via a GDH pathway to ensure proper larval metabolism [20,21]. Methionine-supplemented diets reduce survival but have no effect on reproduction [22], whereas reduced methionine intake extends lifespan and reduces reproduction in low AA status [23,24]. Threonine, another EAA, is a sleep-promoting molecule that links neuronal metabolism to GABAergic control of sleep. Thus, threonine may be the neuronal substrate for sleep homeostasis [9].

In the adult *Drosophila* brain, the mushroom bodies (MBs), an integrative brain center, are a critical structure for appetitive olfactory learning and memory [25,26,27,28], and aversive olfactory memory [29,30]. In addition, the integration of metabolic cues by the MBs has a critical impact on the control of learned behaviors [31,32,33]. The MBs consist of 2000 to 2500 neurons per hemisphere [34], called Kenyon cells [35], which project their axons into two (α and α′) vertical lobes and three (β, β′, and ɣ) medial lobes [34,36]. Kenyon cells receive olfactory inputs from the antennal lobes via projection neurons, process the information, and enable olfactory associative learning [37] and memory [38]. Numerous other stimuli, such as visual, thermosensory, and gustatory inputs [39,40,41,42,43,44], are delivered to the MBs, placing this structure at the center of signal integration and behaviors. In addition, the MB α/β lobes are associated with lifespan [45], which is directly influenced by the quality and quantity of the diet and EAA intake [45].

AA transporters, particularly members of the Solute Carrier family (SLC), play a fundamental role in AA transport in cells, including neurons [46,47,48]. SLC transporters are divided into several families, particularly the SLC7A family which is divided into two subfamilies: the Cationic Amino-acid Transporters (CATs) [49], and the L-type Amino-acid Transporters (LATs) [50]. In *Drosophila*, a specific member of the CAT family, named Slimfast, is expressed in larval fat body cells and in adult DANs, allowing the activation of TOR [51,52] and the AA sensor GCN2 [18], respectively. This activation leads to the regulation of growth and food intake [18,51]. While CATs, such as Slimfast, function as monomers, LATs form Heterodimeric Amino-acid Transporters (HATs) [50,53]. These HATs consist of two subunits, a heavy chain (SLC3A2/CD98hc) and a light chain [54], encoded by five putative genes in *Drosophila* (*minidiscs*, *JhI-21*, *genderblind*, *sobremesa* and *CG1607*) [55,56,57,58]. In mammals, SLC3A2/CD98/4F2hc targets the complex to the plasma membrane, and the light chain determines the specificity of the AA transporter [59,60]. In *Drosophila* S2 cells, a light chain called Minidiscs (MND) is required for leucine transport [61].

In this study, we investigate the function of MND in the response to AAs in the adult brain. First, we show that *mnd* is expressed in both glial cells and neurons in the adult *Drosophila* brain. By generating specific Gal4 and LexA reporter transgenes, we identified two distinct regulatory regions in the *mnd* gene that control expression either in the mushroom body neurons (α/β and ɣ lobe neurons of the MBs) or in cortex glia. We used the MB-specific *mnd* driver to study the neuronal activity in response to different AAs in ex vivo functional brain imaging and revealed that MB neurons respond to a wide range of AAs. Here, we show that *mnd* knockdown in α/β MB lobes results in impaired responses to several AAs, including Leu, Ile, Arg, Asp, Glu, Lys, and Thr. Finally, we found that Leucine, an EAA, activates the MBs through the TOR pathway rather than a GDH signaling pathway. Our data establish that MBs are an important brain center for internal AA sensing that depends on the presence of MND, and this highlights how amino acid transporters, such as SLC family transporters, can influence neuronal activity, which may provide new clues to better understand the regulation of neuronal activity.

## 2. Materials and Methods

### 2.1. Drosophila Strains

All *Drosophila melanogaster* strains were maintained on standard cornmeal/yeast/agar medium, at 25 °C, on a 12 h:12 h light–dark cycle, with 50–60% relative humidity. The fly strains used in this study were as follows: *w^1118^, mnd^24−1^-Gal4, mnd^25−1^-Gal4*, and *mnd^24−1^-LexA* (in this study), *UAS-mCD8::GFP* (stock #32186, stock #32193, Bloomington Drosophila Stock Center (BDSC))*, elav-Gal4* (stock #8760, BDSC)*, repo-Gal4* (stock #7415, BDSC)*, c739-Gal4* (stock #7362, BDSC), *c305a-Gal4* (stock #30829, BDSC)*, H24-Gal4* (stock #51632, BDSC)*, OK107-Gal4* (stock #854, BDSC), *UASfrtSTOPfrtmCD8::GFP* (stock #30125, BDSC), *LexAop2-FLP* (stock #55819, BDSC)*, UAS-GCaMP6s* (stock #42746, BDSC), *UAS-mnd^dsRNAkk^* (stock #110217, Vienna Drosophila Stock Research (VDRC)), *UAS-GDH^dsRNAkk^* (stock # 109499, VDRC), *UAS-TOR^TED^* (stock #7013, BDSC), and *UAS-TOR^dsRNA^* (stock #34639, BDSC).

### 2.2. Transgenic Flies Generated

The *mnd^24−1^* regulatory sequence (1346 pb) and the *mnd^25−1^* regulatory sequence (2581 pb) were amplified by PCR from genomic DNA (*w^1118^* strain) with the primers 5′-GGTACCGGTGAGTGCTCCAGTGGTAAA-3′ (forward) and 5′-GGTACCTACCCATTCGCACTGATAACC-3′ (reverse) for *mnd^24−1^* and 5′-GGGTAACGGTTCTCCCTCTATC-3′ (forward) and 5′-CGTTTGAGTCCACATGGTTTTA-3′ (reverse), and were cloned into the pGEM^®^-T Easy vector (Promega, France) for sequencing (Eurofins mwg operon, Germany). The DNA fragments were inserted into the pattB-Gal4-Rev [62], or pattB-LexA-Rev [63], according to the manufacturer’s instructions, to generate pattB-*mnd^24−1^*-Gal4, pattB-*mnd^25−1^*-Gal4, and pattB-*mnd^24−1^*-LexA constructs. The constructs were integrated into the *Drosophila* genome by targeted injection into embryos (Genetic Services INC., Bucharest, Romania). The pattB-*mnd^24−1^*-Gal4, pattB-*mnd^25−1^*-Gal4, and the pattB-*mnd^24−1^*-LexA were integrated at the attP2 site (3rd chromosome) by ΦC31 integrase. Molecular details are available upon request.

### 2.3. gDNA Extraction

Adult flies were frozen and ground in a buffer (0.1 M EDTA pH 8; 0.1 M Tris-HCl pH 9; SDS 1%), incubated for 30 min at 70 °C, and then kept on ice for 30 min after the addition of 28 µL potassium acetate solution 8 M. After centrifugation at 14,000× *g* for 10 min at 4 °C, isopropanol was added to precipitate nucleic acids. After 10 min at −80 °C and centrifugation at 14,000× *g* for 5 min at 4 °C, gDNA was washed twice with 70% ethanol and dissolved in Tris EDTA pH8.

### 2.4. RNA Extraction and RT-PCR

RNAs was extracted from the heads or bodies of 3-day-old *w^1118^* flies using the TRIzol reagent (Invitrogen, Waltham, MA, USA) and treated with RNasefree DNase I (1 U/mL, Thermo scientific, MA, USA) to eliminate contamination from genomic DNA. Total RNA (1 µg) was reverse transcribed using the iScript cDNA Synthesis kit (Bio-Rad, Hercules, CA, USA). PCR reactions were performed using a thermocycler (Bio-Rad). Amplification of cDNA was performed with 5′-TGGGTAACGGTTCTCCCTCT-3′ (forward) and 5′-TTTCGGTACGGATCCTTGAG-3′ (reverse) before *mnd-RC-* and *mnd-RD-*specific PCR experiments. PCR primers were designed for different exons of the *mnd* coding region (see Figure 1a): *mnd-common:* 5′-GGACAATCCCTCATCGTTTG-3′ (forward), 5′-CCTGATTTGGGTATCATCGTG-3′ (reverse); *mnd-RA*: 5′-GTGCGTTCCATCCGATATTC-3′ (forward), 5′-TCCACGTGGTTCTATCATGTTC-3′ (reverse); *mnd-RC*: 5′-TGGGTAACGGTTCTCCCTCT-3′ (forward), 5′-AGCGTGGAGCTTCCAATACA-3′ (reverse); *mnd-RD*: 5′-TTCTCCCTCTATCGGAACCA-3′ (forward), and 5′-GGAGGAACCTCGAACACCT-3′ (reverse). PCR products were visualized by electrophoresis on 3% agarose gel. The 100 pb DNA ladder (Stock #03B-0713, Euromedex, Souffelweyersheim, France) was used.

### 2.5. qRT-PCR

RNA was extracted from 20 heads/genotypes and cDNA was synthesized from RNA by RT-PCR as described above. A standard protocol was used for real-time PCR with FastSYBR^TM^ Green Master Mix (#4385612, Applied Biosystem, ThermoFisher Scientific, Waltham, MA, USA). PCR primers for *mnd:*5′-GGACAATCCCTCATCGTTTG-3′ (forward), 5′-CCTGATTTGGGTATCATCGTG-3′ (reverse); for housekeeping gene *rp49*: 5′-AGGCCCAAGATCGTGAAGAA-3′ (forward), and 5′-TCGATACCCTTGGGCTTGC-3′ (reverse).

### 2.6. Immunohistology

The primary antibodies used were rabbit anti-MND (1:250; [20]) mouse anti-GFP (1:200, A-11120, Invitrogen), rabbit anti-GFP (1:500, A-6455, Invitrogen), and mouse anti-nc82 (1:10, DSHB). The secondary antibodies used were goat anti-mouse IgG Alexa Fluor 488 (A-11029, Invitrogen), goat anti-rabbit IgG Alexa Fluor 488 (A-11008, Invitrogen), goat anti-mouse IgG Alexa Fluor 594 (A-11005, Invitrogen), anti-rabbit IgG Alexa Fluor 594 (A-11037, Invitrogen), and anti-rabbit IgG Alexa Fluor 647 (A-32733, Invitrogen) at 1:400 dilution.

### 2.7. Whole-Mount Immunostaining

Adult brains were dissected in PBS, fixed in 4% paraformaldehyde for 45 min at room temperature (RT), and washed for 6 × 10 min in PBS + 0.3% Triton X-100 (PBS-T) and 1 × 10 min in PBS + 1%Triton X-100 to permeabilize membranes. Tissues were blocked for 1 h at RT in PBS-T containing 10% normal goat serum (NGS; Sigma #G9023). Appropriate primary antibodies were diluted in PBS-T + 5% NGS and incubated with the tissues for 1 day at 4 °C. After washing 6 × 10 min in PBS-T, samples were labeled with the appropriate secondary antibody at 1:400 in PBS-T containing 5% NGS for 3 h at RT. They were washed for 6 × 10 min in PBS and mounted in VECTASHIELD^®^ mounting medium (H-1000, Vector Laboratories, Peterborough, UK). Fluorescence was observed with a Leica TCS SP8 confocal microscope.

### 2.8. Cross-Sectioning of Adult Brains

For sectioning, the proboscis, wings, and legs were removed from adult *Drosophila* bodies and fixed in 4% paraformaldehyde (PFA) in PBS pH 7.2 for 3 h at 4 °C. The fixative was then replaced with 25% sucrose in Drosophila Ringer’s solution (46 mM NaCl, 182 mM KCl, 3 mM CaCl_2_, 10 mM Tris-HCl, pH 7.2) and incubated overnight at 4 °C. The heads were excised, embedded in Tissue-Tek (Sakura Finetek, Torrance, CA, USA), frozen in liquid nitrogen, and sectioned at 14 µm using a cryostat (Leica CM 3050). Sections were washed 2 × 10 min with TBS + 2.5% Triton (TBS-T) and blocked with 5% NGS for 30 min at RT. Primary antibodies were diluted in a blocking solution and incubated with the samples for 1 day at 4 °C. Sections were washed 2 × 10 min with TBS-T and incubated with the appropriate secondary antibodies diluted in a blocking solution and incubated with the samples overnight at 4 °C. Sections were washed 2 × 10 min at RT and mounted in VECTASHIELD^®^ Hardest^TM^ mounting medium. Fluorescence was observed with a Leica TCS SP8 confocal microscope.

### 2.9. Fluorescence Quantification

Immunohistochemical analysis of the MND protein level within the Kenyon cells of control brains (*mnd^24−1^-Gal4* > *UAS-mCD8::GFP*) and *mnd* downregulated brains (*mnd^24−1^-Gal4* > *UAS-mCD8::GFP*) was performed. Confocal images were obtained using a 63× objective using a 1 µm step size, and the same laser power and scanning settings were used for all samples, using a Leica TCS SP8 confocal microscope. Mean MND fluorescence intensity in the Kenyon cells was quantified from confocal z-stack images using FIJI software (ImageJ 1.47 k). The signal from an adjacent region to the Kenyon cells served as an autofluorescence background and was subtracted from the mean MND fluorescence in these neurons. To compare the two genotypes, MND values of control brains (*mnd^24−1^*-Gal4 > *UAS-mCD8::GFP*) were used as a reference (immunofluorescence = 1).

### 2.10. Calcium Imaging

For ex vivo live calcium imaging experiments, 3-day-old adults were immobilized on ice and the brains were carefully and rapidly removed from the capsule head in Ringer’s saline (130 mM NaCl, 5 mM KCl, 2 mM CaCl_2_, 36 mM saccharose, 5 mM HEPES, pH 7.3) and placed on a silicone plate in a 150 µL drop of Ringer’s saline. To prevent any movement during the imaging, the brains were fixed to the silicone support with insect pins in the optic lobes. GCaMP6s fluorescence was recorded using a Leica DM6000B microscope under a 25× water objective. GCaMP6s was excited using a Lumencor light engine supplied with diodes at 485 ± 25 nm. The emitted light was collected through a 505–530 nm bandpass filter. Leica MM AF 2.2.0 was used for data collection and acquisition. Images were acquired at 500 ms per frame at 256 × 256 resolution using an Orca-Flash 4.0 camera. A minimum of 480 images were acquired for each experiment: 100 before and 380 after the AA application. The first ten frames before the AA application were used to establish the baseline F_0_. Regions adjacent to the region of interest were used to determine the autofluorescence background level. Changes in fluorescence from initial fluorescence (ΔF/F_0_) were calculated as the peak fluorescence after t = 100 frames minus F_0_ versus F_0_, [64]. AA solutions are extemporaneously prepared with an appropriate concentration in Ringer’s saline. All L-AAs used in this study were purchased from Sigma–Aldrich.

### 2.11. Statistical Analysis

Prism 9 software (Graphpad, v9.0, Boston, MA, USA) was used for statistical analyses. Shapiro–Wilk and D’Agostino–Pearson tests were used to assess normality for all individual experiments. The two-tailed unpaired Student’s t-test was used for comparison between two groups with normally distributed data, and the Mann–Whitney test was used for data that did not pass the normality test. For comparisons between more than two groups with normally distributed data, one-way ANOVA was used, and the Kruskall–Wallis test was used for data that did not pass the normality test.

## 3. Results

### 3.1. mnd Is Expressed in the Adult Drosophila Brain

We first examined the expression pattern of *mnd* in adult *Drosophila*. Three *mnd* transcripts were predicted. All should derive through alternative splicing of the 5′UTR, generate *mnd-RA*, *mnd-RC,* and *mnd-RD*, which all encode the same protein [55], see Figure 1a. According to the Flybase prediction (FB2024_03, released 25 June 2024), *mnd* may be expressed in different tissues including the adult brain. To test whether these splice forms may be present in different tissues, we first attempted to detect these three *mnd* transcripts by RT-PCR in adult heads and bodies. We designed specific primers for each cDNA of *mnd* transcripts, represented by colored arrows in Figure 1a. After separating the RT-PCR fragments by agarose gel electrophoresis, we found that *mnd-RA* (111 pb) and *mnd-RC* (183 pb) are expressed in the heads and the bodies, while *mnd-RD* (114 pb) is not (Figure 1b), indicating that *mnd-RD* may be very weakly expressed or does not exist as predicted. The amplified fragment for each mRNA has been sequenced and confirmed the expression of the three different transcripts.

Next, we also investigated MND expression using MND antibodies [20], and we showed that MND is broadly expressed in the brain (Appendix A) and that the antibodies are specific to MND (Appendix A). To identify the type of cells expressing MND, we used GFP-expression driven by a neuronal-specific driver (*elav-Gal4*) or a glia-specific driver (*repo-Gal4*), the two major cell types in the brain. MND was detected in both neurons and glial cells (Figure 1c).

### 3.2. The Regulatory Promoter Sequences of Mnd Specifically Drives Expression either in Neurons or in Glial Cells

We then attempted to identify *mnd* regulatory sequences capable of driving expression in the nervous system by generating reporter transgenic lines. By bioinformatic analysis, we found two potential regulatory sequences illustrated by the blue box for the region called *mnd^24−1^* and the green box for the *mnd^25−1^* region in Figure 2a.

We generated a *Gal4* driver transgene, *mnd^25−1^-Gal4,* containing the regulatory sequence of *mnd* corresponding to the 5′UTR region of the second *mnd* exon (Figure 2a). This transgene drives expression in the adult brain, particularly in glial cells (Figure 2b).

We then found that the *mnd^24−1^-Gal4* transgene containing the regulatory sequences of the 5′ region of the first *mnd* exon (Figure 2a) drives expression in the brain, particularly in the neurons of the MBs (Figure 2(c1)). Specifically, we observed that *mnd^24−1^-Gal4* driver expression in the MBs appeared to be restricted to certain lobes. We performed immunostaining on *mnd^24−1^-Gal4 > UAS-mCD8::GFP* brains to examine the presence of MND (Figure 2(c1″,c1**)). We observed that MND and GFP co-localized in the cell body of the Kenyon cells (Kcs) (Figure 2(c1*,c1**,c1***)) and found weak MND expression in the MB calyx but not in MB lobes (Figure 2(c1′,c1″,c1″′)). This supports that the neuronal expression of MND is under the control of the specific *mnd^24−1^* regulatory sequence.

The *mnd^24−1^-LexA* driver, which contains the same regulatory sequence, results in a slightly more widespread GFP expression throughout the brain, particularly in the MB neurons and the optic lobes (Figure 2(c2)). MND immunostaining on *mnd^24−1^-LexA > LexAop-mCD8::GFP* brains revealed that MND and GFP co-localized in the cell bodies of the Kenyon cells (Kcs) (Figure 2(c1′,c1″,c1″′)), and we found weak MND expression in the MB calyx but not in MB lobes (Figure 2(c2′,c2″,c2″′)), such as *mnd^24−1^-Gal4* driver.

These tools allowed us to perform intersectional genetic strategies using the flippase/FRT system to observe cells simultaneously positive for *mnd^24−1^-Gal4* and *mnd^24−1^-LexA*. In the brains of *mnd^24−1^-Gal4/mnd^24−1^-LexA, UAS > stop > mCD8::GFP*; *LexAop-FLP* flies, we detected GFP and MND expression in MB neurons (Figure 2(c3)), confirming that *mnd^24−1^-Gal4* and *mnd^24−1^-LexA* drivers can mimic part of MND expression in the brain. MND presence was confirmed by MND labeling and co-localization with GFP signal in KCs (Figure 2(c3′,c3″,c3″′)), but not in MB lobes (Figure 2(c3′,c3″,c3″′)), consistent with what we observed with *mnd^24−1^-Gal4* and *mnd^24−1^-LexA* individually. Altogether, these results demonstrate that *mnd* expression in mushroom body neurons is dependent on the *mnd^24−1^* regulatory sequence.

The MBs are constituted by about 2000 and 2500 Kenyon cells per hemisphere, which send their axons in different directions forming three different lobes, the α/β, α′/β′, and ɣ lobes, in the anterior part of the brain [34,44]. Each MB lobe processes specific sensory inputs and thus drives particular behavioral outcomes [65]. To identify the lobes in which *mnd^24−1^* is expressed, we used an intersectional genetic strategy combining *mnd^24−1^-LexA > LexAop-FLP*, and different *MB-lobe-specific-Gal4* > *UAS* > *stop* > *mCD8::GFP*, which allowed us to identify common cells expressing *mnd^24−1^-LexA* and each *MB-lobe-specific-Gal4* drivers.

The GFP expression could be seen in brains when mnd*^24−1^-LexA* (Figure 3e) and the α/β lobe specific driver (*c739-Gal4)* (Figure 3(a2)) were used, indicating that the *mnd^24−1^-LexA* drives expression in the α/β lobes (Figure 3(a2e)), and MND immunostaining co-localized with GFP, confirming the presence of MND inside these neurons on *c739-Gal4 > UAS-mCD8::GFP* brains (Figure 3(a1,a1′,a1″)). By contrast, combining *mnd^24−1^-LexA* (Figure 3e) and the α′/β′ lobe specific driver *(c305-Gal4)* (Figure 3(b2)) did not result in GFP expression, suggesting that *mnd^24−1^-LexA* driver does not activate expression in the α′/β′ lobes (Figure 3(b2e)). Surprisingly, MND labeling co-localized with GFP on *c305-Gal4 > UAS-mCD8::GFP* brains (Figure 3(b1,b1′,b1″)), suggesting that MND is expressed in α′/β′ lobes, but is not driven by the *mnd^24−1^* regulatory sequence. At least at the intersection between *mnd^24−1^-LexA* and the ɣ lobe specific driver (*H24-Gal4)* (Figure 3(c2)), we observed GFP expression in *mnd^24−1^* and *H24* expressing neurons, indicating that the *mnd^24−1^-LexA* driver activates expression in the ɣ lobes (Figure 3(c2e)), and MND immunostaining co-localized with GFP, confirming the presence of MND inside these neurons in *H24-Gal4* > *UAS-mCD8::GFP* brains (Figure 3(c1,c1′,c1″)). We also performed genetic intersectional experiments between *mnd^24−1^-LexA* and *OK107-Gal4* (Figure 3(d2)), which allowed us to confirm that the *mnd^24−1^-LexA* driver is expressed in α/β and ɣ lobes and not in α′/β′ lobes of the MBs (Figure 3(d2)). The presence of MND was confirmed by co-localization of MND and GFP (Figure 3(d1,d1′,d1″)). Thus, we have shown that the *mnd^24−1^* promoter sequence specifically drives expression in the α/β lobes and in the ɣ lobes.

### 3.3. MND Is Required for AA Dependent Activity of Kenyon Cells

MND is present in MB neurons, but its precise function as an AA transporter has not yet been demonstrated in vivo. MND has been described as a Leu transporter in S2 cell cultures [61] and as a Leucine sensor in larval IPCs where it enables DILP release [20]. To test whether and how MND impacts neuronal activity within the MB, we expressed the calcium sensor *GCaMP6s* in *mnd^24−1^-Gal4* expressing neurons (*mnd^24−1^-Gal4* > *GCaMP6s*) and studied their activation using calcium imaging on ex vivo brains. In the brain, glutamate released by neurons or glial cells activates MB neurons, which express different glutamate receptors [66,67,68,69,70]. Therefore, we decided to test the activity of the *mnd^24−1^*-positive MB neurons in response to different concentrations of Glutamate in 3-, 6-, or 9-day-old flies. We tested Glu concentrations from 0.2 mM to 20 mM, which was previously used to activate neurons in *Drosophila* [71,72,73,74,75]). The highest response was observed in 3-day-old flies when 20 mM Glu was applied to the brains (Appendix A). Consequently, we tested the response to the remaining 19 natural L-AAs using the same conditions.

Our results show that the *mnd^24−1^-Gal4* > *GCaMP6s* expressing MB neurons that form the α/β lobes can respond to all L-AAs except Serine and Tryptophan (Figure 4a,c). We noticed that three amino acids, namely, Glutamate, Proline, and Threonine, elicited the highest responses among all L-AA tested (Figure 4c).

To elucidate the role of MND in MB activity in response to L-amino acids, we examined the activity of *mnd^24−1^-Gal4* > *GCaMP6s*;*mnd^dsRNAKK^* brains in which *mnd* is knocked-down only in the *mnd^24−1^* positive neurons expressing the calcium sensor GCaMP6s (Figure 4b,c and Appendix A). These results are consistent with our previous experiments showing that MND is involved in the transport of Leu, Ile, and not Val in larval IPCs to enable the release of DILPs [20]. In addition, we show that MND is also involved in the sensing of other L-AAs such as Arg, Asp, Glu, Lys, and Thr in the α/β MB neurons (Figure 4c).

### 3.4. The TOR Signaling Pathway Mediates the Stimulation of the MBs by Leucine

Leucine, an EAA supplied through dietary food intake, is capable of regulating cell function through either the Target Of Rapamycin (TOR) pathway or the Glutamate Dehydrogenase (GDH) pathway [20,76,77,78,79]. In a previous study, we demonstrated that Leu triggers the release of DILPs via the GDH pathway in an MND-dependent manner [20]. Since Leu and Ile, two amino acids, activate the MBs that express *mnd* (Figure 4c), we examined the downstream GDH and TOR signaling pathways (Figure 5a). In adult brains, inactivation of GDH did not reduce the activity of the α/β MB neurons in response to L-Leu (Figure 5b). We next investigated whether MB activation by L-Leu could be mediated by the TOR pathway. Overexpression of a dominant negative form of TOR (TOR^TED^) in *mnd^24−1^*-positive neurons impaired the L-Leu-dependent stimulation of the MBs (Figure 5b). Similarly, the *mnd^24−1^*-positive MB neurons are not activated by L-Leu when TOR expression is silenced by RNAi (Figure 5b). These results show that the stimulation of the α/β MB neurons by Leucine depends on MND and the TOR pathway, but not on the GDH pathway.

## 4. Discussion

### 4.1. mnd Is Expressed in Different Cell Types in the Adult Brain

In this study, we show that MND, a LAT1-like AA transporter, is present in the adult *Drosophila* brain. Our data reveal the expression of two *mnd* mRNAs, *mnd-RA* and *mnd-RC,* within the *Drosophila* head, while the third predicted *mnd-RD* transcript was undetectable, indicating either its absence or very low expression levels. Furthermore, we have shown that MND is expressed by both neurons and glia, suggesting that *mnd* may be regulated by distinct regulatory sequences in the *mnd* promoter. By examining the promoter regulatory region of *mnd*, we show that two specific regulatory sequences drive expression in two different regions of the adult brain. *mnd^25−1^-Gal4* drives MND expression specifically in cortex glial cells. These findings are consistent with previous studies showing that *mnd* is expressed in glial cells [75,80]. We observe that *mnd^24−1^-Gal4* is expressed in MB neurons. Our intersectional genetic strategies reveal that the *mnd^24−1^* regulatory sequence drives expression in the α/β and ɣ lobes but not in the α′/β′ lobes of the MBs. Immunostaining of MND showed that MND is located in the neurons of the α/β and ɣ lobes but also in the α′/β′ lobes of the MBs, suggesting the *mnd^24−1^* regulatory sequence only controls the expression of *mnd* in a subset of neurons of the MBs and implying that another neuronal regulatory sequence of *mnd* exists.

### 4.2. AAs Stimulate Neuronal Activity in the MBs

The EAAs Leu, Ile, and Thr must be supplied by the diet and they can act as nutrient signals to regulate the state of nutritional homeostasis [9,20,81]. Interestingly, the MBs are an integrated center for hunger control of food-seeking behaviour receiving input signals of hunger and satiety [33]. DANs regulate MB activity and control innate olfactory behaviour. These DANs are under the control of satiety signals such as insulin-like peptides and AstA or hunger signals such as NPF, sNPF, and serotonin [33]. Here, we show that the α/β lobes of the MBs are activated by EAAs such as Leu, Ile, and Thr. Thus, EAAs, nutrients arriving from the gut, can directly or indirectly affect neuronal functions driven by the MBs. We also show that Glu, an excitatory neurotransmitter, significantly enhances the activity of the MBs. The increase in calcium activity in the MBs induced by Glu may be due to a direct or indirect action of Glu.

### 4.3. Disruption of mnd Impairs Response of the MBs to AAs

Initially, *mnd* was described to be involved in the non-autonomous development of imaginal discs in the larvae [55]. We did not observe any changes in the shape and length of the α/β lobes when *mnd* was downregulated in the MBs, suggesting that MND is not involved in the development of the MBs [82]. In our experiments, the downregulation of *mnd* in the MBs does not reduce neuronal activity in response to amino acids such as Ala, Asp, Gly, His, Phe, Pro, Tyr, and Val (Figure 4c). Disruption of MND function in the α/β lobes of the MBs reduces neuronal activity in the response to certain AAs, such as Leu and Ile. In mammals, Leucine regulates the activity of hypothalamic POMC neurons to control food intake [83] through mTOR signaling [84]. In *Drosophila*, MBs integrate hunger and food signals to monitor food-seeking behaviour [33]. We show that MBs are sensitive to Leu suggesting that MBs integrate protein satiety signals to adjust their activity. Previously, we have shown that MND is required for the sensing of Leu in the larval brain IPCs, resulting in an increase of neuronal activity and, DILPs release through a GDH pathway [20,21]. Surprisingly, in adult MBs, Leucine regulates neuronal activity through the TOR pathway but not the GDH pathway (Figure 6). Interestingly, MND-expressing glial cells show no calcium activity in response to Leucine (Appendix A), suggesting a different function of MND in this specific cell type or the cellular pathway is calcium-independent.

Leucine affects calcium activity in the α/β lobes of MBs through MND, a LAT1-like amino acid transporter, via the TOR pathway. The activity of L-AAs such as Arg, Asp, Glu, Leu, Ile, Lys, and Thr on the MBs is MND-dependent, and MND could potentially affect AA receptors or transporters or the downstream pathways.

In summary, our work shows that extracellular AAs can induce calcium activity in the MB α/β lobes and that, for certain AAs, this activity is MND-dependent. MND, a LAT1-like transporter, may have a direct or indirect effect on AA receptors or transporters in the MBs. In the brain, Leucine is sensed by glial cells to promote a preference for Leucine-containing food [85]. Furthermore, ensheating glial cells surrounding the MBs release Glutamate onto MBs required for associative memories [70]. Since MND is also expressed in glial cells, it could be interesting to investigate its potential role in AAs transport in these cells. Further investigation of the cellular and molecular mechanisms will provide new insights into the role of MND in MBs and glial cells for proper regulation of brain activity.

## Figures and Tables

**Figure 1 cells-13-01340-f001:**
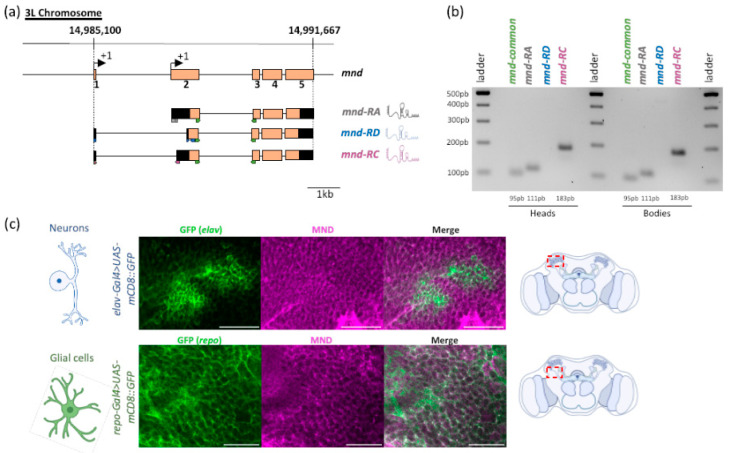
*mnd* is expressed in the adult *Drosophila* brain. (**a**) Schematic representation of the structure of the *mnd* gene, located on the left arm of the third chromosome (14,985,100–14,991,667 bp). Exons from 1 to 5 are represented by orange boxes and introns by black lines. Three mRNA isoforms were predicted by the ORF analysis from expasy.org: *mnd-RD* (in blue), *mnd-RC* (in purple), and *mnd-RA* (in gray). The 5′ and 3′ untranslated regions are indicated by black boxes. The arrows under the transcripts indicate the primers used for the RT-PCR analysis: green for the common region of all three alternative *mnd* mRNAs; gray for *mnd-RA*; purple for *mnd-RC*; and blue for *mnd-RD.* Scale bar, 1 kb. (**b**) *mnd* expression analyzed by RT-PCR using RNAs extracted from *w^1118^* adult heads and bodies. PCR products were analyzed by electrophoresis on agarose gel. *mnd-RC* (183 pb) and *mnd-RA* (111 pb) are both present in the adult heads and bodies, whereas *mnd-RD* (114 pb) was not detected in heads or bodies. Primers used for RT-PCR are indicated in (**a**) with colored arrows: green for the common portion of the three alternative *mnd* mRNAs; gray for *mnd-RA*; purple for *mnd-RC*; and blue for *mnd-RD.* (**c**) Representative images of double immunostaining with anti-MND (magenta) and anti-GFP (green) on *elav-Gal4 > UAS-mCD8::GFP* brain (neuronal marker) or on *repo-Gal4 > UAS-mCD8::GFP* brain (glial cell marker). Red boxes illustrate the area of the MB calyx (upper image) and the cortex glia (lower image) in the brain where the images were recorded. A total of 12 brains for each condition were examined. MND is present in both neurons and glial cells in the adult brain. White indicates the overlap of the two markers on merged images. Scale bar, 50 µm.

**Figure 2 cells-13-01340-f002:**
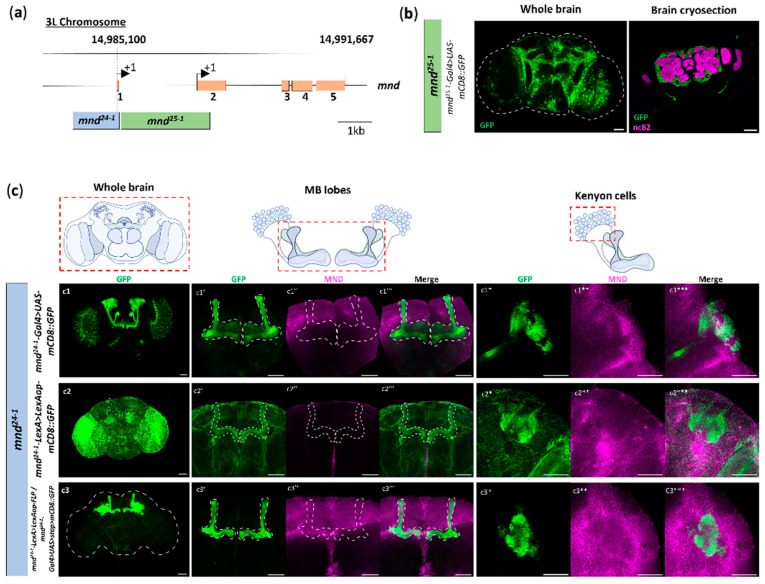
*mnd* regulatory sequences lead to different expression patterns of MND. *mnd* regulatory sequences lead to different MND expression patterns, and *mnd^24−1^* tools mimic a part of *mnd* expression in the MBs. (**a**) Schematic representation of the structure of the *mnd* gene. Exons from 1 to 5 are represented by orange boxes and introns by black lines. The upstream regulatory sequences of the *mnd* gene are represented by the blue and the green box and are designed as *mnd^24−1^* and *mnd^25−1^*, respectively, to generate transgenic driver lines. *mnd^24−1^* is located on the first exon of *mnd* with the first initial site of transcription, and *mnd^25−1^* is located on the second exon of *mnd* including the second initial site of transcription. Scale bar, 1 kb. (**b**) *mnd^25−1^-Gal4* induces mCD8::GFP expression in the cortex glia. Representative image of mCD8::GFP expression driven by *mnd^25−1^-Gal4* (*mnd^25−1^-Gal4 > UAS-mCD8::GFP*) in the whole brain magnified by anti-GFP immunostaining, and in brain cryosection labeled by anti-GFP and anti-nc82 to visualize neuropiles. Eight brains were examined. (**c**) Collection of representative Z-projections and images of mCD8::GFP driven by *mnd^24−1^* tools and MND labeling. (**c1**) *mnd^24−1^-Gal4* induces mCD8::GFP expression in the mushroom body lobes. Representative Z-projections of mCD8::GFP expression driven by *mnd^24−1^-Gal4* (*mnd^24−1^-Gal4 > UAS-mCD8::GFP*) in the whole brain, and magnified by anti-GFP immunostaining (**c1**). Representative Z-projections of double immunostaining in MB lobes: anti-GFP (**c1′**), anti-MND (**c1″**), and merge (**c1″′**). Representative images of mCD8::GFP and MND showing the colocalization in Kenyon cells: anti-GFP (**c1***), anti-MND (**c1****), and merge (**c1*****). A total of 34 brains were examined. (**c2**) *mnd^24−1^-LexA* induces mCD8::GFP expression in the brain. Representative Z-projection of mCD8::GFP expression driven by *mnd^24−1^-LexA* (*mnd^24−1^-LexA > LexAop-mCD8::GFP*) in the whole brain, and magnified by anti-GFP immunostaining (**c2**), in MB lobes anti-GFP (**c2′**), anti-MND (**c2″**), and merge (**c2″′**). Representative images of mCD8::GFP and MND showing the colocalization in Kenyon cells: anti-GFP (**c2***), anti-MND (**c2****), and merge (**c2*****). A total of 25 brains were examined. (**c3**) Representative Z-projections of mCD8::GFP expression resulting from the genetic intersection between *mnd^24−1^-Gal4* and *mnd^24−1^-LexA* drivers (*mnd^24−1^-Gal4 > UAS > stop > mCD8::GFP/mnd^24−1^-LexA > FLP*), and magnified by anti-GFP immunostaining (**c3**). Both the *mnd^24−1^-Gal4* and *mnd^24−1^-LexA* drivers are expressed in the MB lobes: anti-GFP (**c3′**), anti-MND (**c3″**), and merge (**c3″′**). Representative images showing colocalization of anti-GFP and anti-MND in Kenyon cells anti-GFP (**c3***), anti-MND (**c3****), and merge (**c3*****). A total of 13 brains were examined. For all images, the scale bar is 50 µm.

**Figure 3 cells-13-01340-f003:**
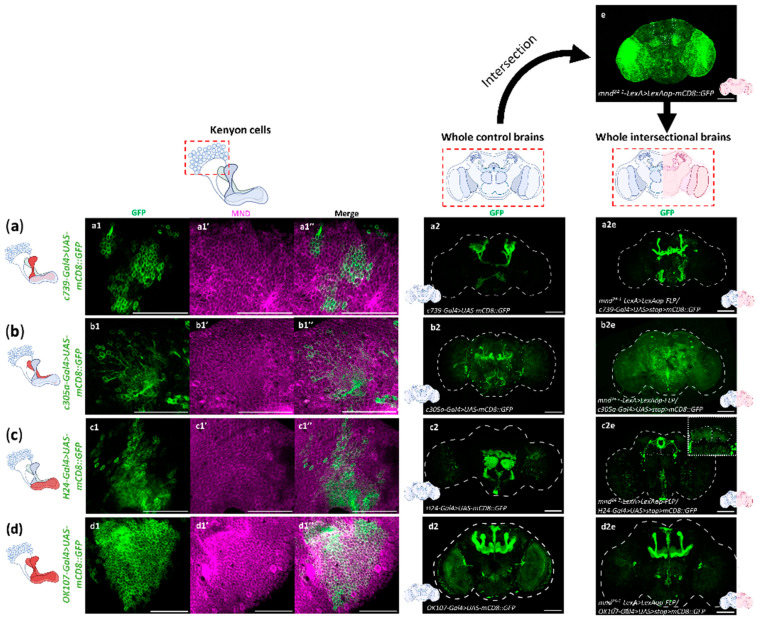
*mnd* is expressed in Kenyon cells forming α/β and ɣ lobes. Genetic intersectional strategy between *mnd^24−1^-LexA* line, which mimics *mnd* expression in the brain, and *MB-lobe-specific-Gal4* driver lines to reveal common cells. (**a**) Genetic intersectional GFP expression between *c739-Gal4*, an α/β lobe specific driver, and *mnd^24−1^-LexA,* and magnified by anti-GFP immunostaining. Representative images of anti-GFP (**a1**) and anti-MND (**a1′**) in Kenyon cells (*c739-Ga4* > *UAS-mCD8::GFP*), merge (**a1″**). Representative z-projection of mCD8::GFP expression driven by *c739-Gal4* (*c739-Ga4* > *UAS-mCD8::GFP*) in whole brain (**a2**). Seven brains were examined. Representative z-projection of mCD8::GFP expression driven by *mnd^24−1^-LexA* (*mnd^24−1^-LexA* > *LexAop-mCD8::GFP*) (**e**). A total of 25 brains were examined. Representative z-projection of mCD8::GFP expression resulting from the intersection of *c739-Gal4* and *mnd^24−1^LexA* (*mnd^24−1^-LexA* > *LexAop-FLP/c739-Gal4* > *UAS* > *stop* > *mCD8::GFP*) (**a2e**). A total of 11 brains were examined. *mnd^24−1^-LexA* is expressed in the α/β lobes of the MBs. (**b**) Genetic intersectional GFP expression between *c305a-Gal4*, a specific driver of α′/β′ lobes and *mnd^24−1^-LexA*, and magnified by anti-GFP immunostaining. Representative images of anti-GFP (**b1**) and anti-MND (**b1′**) in Kenyon cells (*c305a-Ga4* > *UAS-mCD8::GFP*), merge (**b1″**). Representative z-projection of mCD8::GFP expression driven by *c305a-Gal4* (*c305a-Ga4* > *UAS-mCD8::GFP*) in whole brain (**b2**). A total of 16 brains were examined. Representative z-projection of mCD8::GFP expression driven by *mnd^24−1^-LexA* (*mnd^24−1^-LexA* > *LexAop-mCD8::GFP*) (**e**). A total of 25 brains were examined. Representative z-projection of mCD8::GFP expression resulting from the intersection of *c305a-Gal4* and *mnd^24−1^-LexA* (*mnd^24−1^-LexA* > *LexAop-FLP/c305a-Gal4* > *UAS* > *stop* > *mCD8::GFP*) (**b2e**). A total of 11 brains were examined. *mnd^24−1^-LexA* is not expressed in the α′/β′ lobes of the MBs. (**c**) Genetic intersectional GFP expression between *H24-Gal4*, a specific driver of ɣ lobes and *mnd^24−1^-LexA*, and magnified by anti-GFP immunostaining. Representative images of anti-GFP (**c1**) and anti-MND (**c1**′) in Kenyon cells (*H24-Ga4* > *UAS-mCD8::GFP*), merge (**c1″**). Representative z-projection of mCD8::GFP expression driven by *H24-Gal4* (*H24-Ga4* > *UAS-mCD8::GFP*) in whole brain (**c2**). A total of 17 brains were examined. Representative z-projection of mCD8::GFP expression driven by *mnd^24−1^-LexA* (*mnd^24−1^-LexA* > *LexAop-mCD8::GFP*) (**e**). A total of 25 brains were examined. Representative z-projection of mCD8::GFP expression resulting from the intersection of *H24-Gal4* and *mnd^24−1^-LexA* (*mnd^24−1^-LexA* > *LexAop-FLP/H24-Gal4* > *UAS > stop* > *mCD8::GFP)* (**c2e**). A total of 12 brains were examined. *mnd^24−1^-LexA* is weakly expressed in the ɣ lobes. (**d**) Genetic intersectional GFP expression between *OK107-Gal4*, a specific driver of all MB lobes, and *mnd^24−1^-LexA*, and magnified by anti-GFP immunostaining. Representative images of anti-GFP (**d1**) and anti-MND (**d1′**) in Kenyon cells (*OK107-Ga4* > *UAS-mCD8::GFP*), merge (**d1″**) Representative z-projection of mCD8::GFP expression driven by *OK107-Gal4* (*OK107-Ga4* > *UAS-mCD8::GFP*) in whole brain (**d2**). A total of 10 brains were examined. Representative z-projection of mCD8::GFP expression driven by *mnd^24−1^-LexA* (*mnd^24−1^-LexA* > *LexAop-mCD8::GFP*) (**e**). A total of 25 brains were examined. Representative z-projection of mCD8::GFP expression resulting from the intersection of *OK107-Gal4* and *mnd^24−1^-LexA (mnd^24−1^-LexA* > *LexAop-FLP/OK107-Gal4* > *UAS* > *stop* > *mCD8::GFP)* (**d2e**). A total of 11 brains were examined. *mnd^24−1^-LexA* is expressed in the α/β and ɣ lobes of the MBs. For all images the scale bar is 50 µm.

**Figure 4 cells-13-01340-f004:**
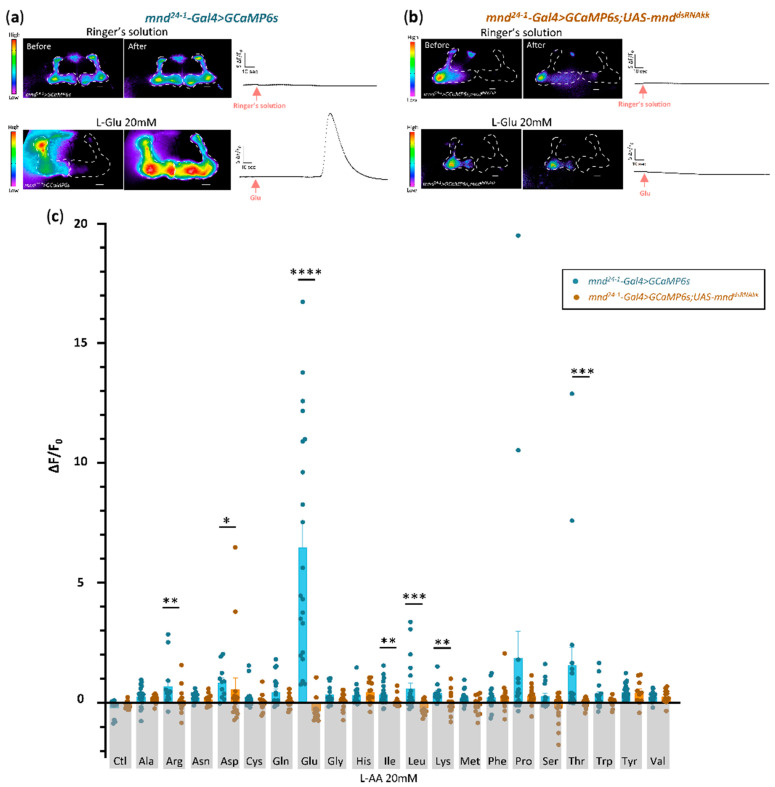
MND is required for AA sensing in the MBs. Real-time calcium imaging of ex vivo brains expressing a calcium sensor in the α/β lobes of the MBs in control brains (*mnd^24−1^-Gal4* > *GCaMP6s*) or in *mnd* downregulated brains (*mnd^24−1^-Gal4* > *GCaMP6s*;*UAS-mnd^dsRNAkk^*) of 3-day-old flies exposed to each L-AA at a concentration of 20 mM. The different intensities of basal GCaMP6s levels in the vicinity of the two lobes are sometimes difficult to convert to similar rainbow colors, and thus in some brains only one lobe appears in false color. (**a**) Representative images, in false colors, showing the fluorescence level before (basal activity) and after the addition of either control Ringer’s solution or Glu (20 mM) in control brains (*mnd^24−1^-Gal4* > *GCaMP6s*). Scale bar, 50 µm. Line plots of fluorescence changes (∆F/F_0_) in α/β lobe neurons stimulated with Ringer’s solution or 20 mM of Glu for one representative brain. Stimulus application is indicated by a red arrow. (**b**) Representative images, in false colors, showing the fluorescence level before (basal activity) and after the addition of either control Ringer’s solution or Glu (20 mM) in *mnd* downregulated brains (*mnd^24−1^-Gal4* > *GCaMP6s*;*UAS-mnd^dsRNAkk^*). Scale bar, 50 µm. Line plots of fluorescence changes (∆F/F_0_) in α/β lobe neurons stimulated with Ringer’s solution or 20 mM of Glu for one representative brain. Stimulus application is indicated by a red arrow. (**c**) Averaged fluorescence intensity of positive or negative peaks ± SEM for control brains (*mnd^24−1^-Gal4* > *GCaMP6s*, blue histograms) and for *mnd* downregulated brains (*mnd^24−1^-Gal4* > *GCaMP6s*;*UAS-mnd^dsRNAkk^*, orange histograms) in response to either Ringer’s solution (Ctl) or a specific L-AA at 20 mM. All individual data are shown by dots (*n* = 10 to 23). For each L-AA, data obtained from *mnd* downregulated brains (*mnd^24−1^-Gal4* > *GCaMP6s*; *UAS-mnd^dsRNAkk^*) were compared to the corresponding control (*mnd^24−1^-Gal4* > *GCaMP6s*) using a Mann–Whitney test. The absence of * for a given AA indicates that the data are not statistically different between the two conditions. *: *p* < 0.05; **: *p* < 0.01; ***: *p* < 0.001; ****: *p* < 0.0001.

**Figure 5 cells-13-01340-f005:**
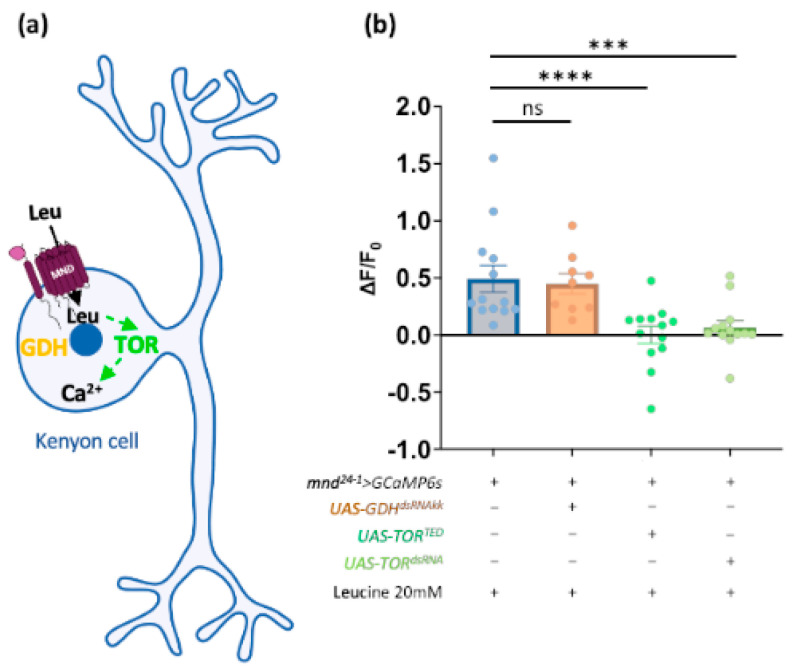
Leucine-mediated activity is TOR-dependent in the MBs. (**a**) Schematic representation of a Kenyon cell and the putative pathways downstream of leucine activity. (**b**) Real-time calcium imaging of ex vivo brains expressing a calcium sensor in the α/β lobes of the MBs in control brains (*mnd^24−1^-Gal4* > *GCaMP6s*), in GDH-downregulated brains (*mnd^24−1^-Gal4* > *GCaMP6s*; *UAS-GDH^dsRNAkk^*), or in TOR downregulated brains (*mnd^24−1^-Gal4* > *GCaMP6s*; *UAS-TOR^TED^* or *mnd^24−1^-Gal4* > *GCaMP6s*; *UAS-TOR^dsRNA^*) of 3-day-old flies exposed to L-Leu at a concentration of 20 mM. Each histogram represents the averaged fluorescence intensity of peaks ± SEM in α/β lobes of the MBs. All individual data are shown by dots (*n* = 9 to 13). All data were compared with the control (*mnd^24−1^-Gal4* > *GCaMP6s*) by a Mann–Whitney test. ns: not significant, ***: *p* < 0.001; ****: *p* < 0.0001.

**Figure 6 cells-13-01340-f006:**
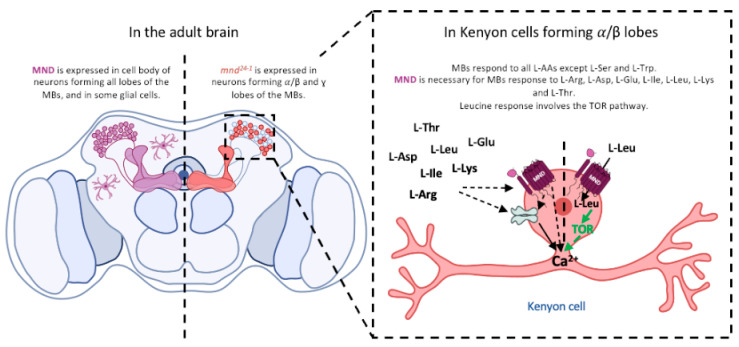
**A** model for AA sensing through MND in MBs.

## Data Availability

The raw data supporting the conclusions of this article will be made available by the authors on request.

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
