# Peer review of "A LAT1-Like Amino Acid Transporter Regulates Neuronal Activity in the *Drosophila* Mushroom Bodies"

_cells, 2024, doi:10.3390/cells13161340_

Round 1

Reviewer 1 Report

Comments and Suggestions for Authors

I have reviewed A LAT1-like amino acid transporter regulates neuronal activity in the Drosophila mushroom bodies by Delescluse et al. The authors first use RT-PCR and convincingly show that two of three predicted splice variants of MND are expressed in adult heads. They show labeling of the brain using a previously generated antibody (see major concerns). They next generate drivers representing predicted regulatory regions of the MND gene and map the expression patterns of the two drivers. It is difficult to determine which neurons are thought to be co-labeled (see major concerns). Regardless of these issues, using an intersectional approach with the two drivers they convincingly label the lobes of the MBs. They then compare the labeling on the MND antibody with GAL4 drivers that are expressed in more specific subsets of cells. The intersectional approach using the MND drivers with additional  GAL4 drivers supports the idea that MND is likely expressed in MBs but the extensive labeling of other regions makes it difficult to determine whether there are any cells in which it is not expressed. In Fig S1 and 4 they use GCAMP to test the response to glutamate and other amino acids. The importance of  the glutamate data are not clear since it is an excitatory neurotransmitter in the fly CNS and many neurons would likely respond to it either directly or if they were downstream of other excitable cells (see minor concerns). The use of other amino acids is more directly tied to their central idea and the data support their conclusion that MND may act as an amino acid transporter in MB neurons.  The use of MND RNAi to block the effects when expressed with the same driver is a critical control and supports the idea that Leu is acting directly on MB cells rather than indirectly affecting other neurons.  Both a dominant negative form of TOR and TOR RNAi also block the response to leucine. As the authors suggest this convincingly shows that stimulation of the ɑ/β MB neurons by Leucine depends on the TOR pathway.

In summary, the paper nicely demonstrates that neurons in the adult brain show a response to a subset of amino acids, complementing the authors previous work in vitro and in larvae. However, the authors should more clearly explain whether they think this is a property of MND in all cells, or all neurons, or all neurons and glia. This would require clarifying the expression data (see below).

Major Concerns

The specificity of the antibody is not clear making it difficult to interpret all of the co-labeling experiments. Higher resolution and/or single optical sections might partially mitigate this problem, but in the current figures it appears that every cell is labeled.  It is certainly possible that MND is expressed in neurons and glia as suggested by the authors, but this should be placed in a broader context. Is it expressed in all cells throughout the brain and body or just neurons and glia?

Similarly, based on the current data it is not clear whether it is expressed in all neurons and glia or just a subset. The expression data support the idea that it is expressed in MB cells, validating the use of these for GCaMP assay. However, their data also appear consistent with expression in many and perhaps all cells within the brain. The antibody appears to label every cell in the field of view making it very difficult to appreciate how it can be effectively used for co-localization experiments. Similarly, the intersectional approach using  the MND  drivers with additional  GAL4 drivers supports the idea that MND is likely expressed in MBs but the extensive labeling of other regions makes it difficult to determine whether there are any cells in which it is not expressed. As such, the paper would be strengthened by emphasizing the use of MB neurons as a reasonable place to test the response of MND to amino acids rather than trying to show that it is specifically expressed in these cells. Indeed, if MND is expressed in other cells, it is not clear why the authors chose to focus only on MBs rather than testing the response of other cells as well and this should be briefly stated.

Minor issues

A large number of cells are likely to respond to glutamate and it is not clear whether the response of the MB neurons is direct. Therefore, the authors might consider treating the use of Glut as simply a positive control for their GCaMP assays rather than trying to explain how it may affect the MBs.

Reviewer 2 Report

Comments and Suggestions for Authors

This is an overall well-written manuscript which characterizes the expression and role of minidiscs (mnd), an amino acid transporter, in the Drosophila brain. The authors use both RT-PCR and immunostaining to demonstrate expression in body and brain regions of the fly. In the brain, they are able to detect MND expression in both neurons and glia, and among neuronal populations, they show MND expression in Kenyon cells of the mushroom body (MB). This was facilitated by transgenic lines generated by the authors that contain mnd regulatory sequences upstream of the Gal4 transcription factor. To verify that mnd functions as an amino acid transporter, the authors measured neuronal activity in MB neurons by looking at a calcium reporter following ex vivo treatment with different amino acids. Neuronal activation after amino acid treatment was indeed dependent on mnd, because knockdown of the gene in the cells in which it is expressed resulted in suppression of the calcium reporter signal. For one amino acid, leucine, the authors also show that activation of mnd-expressing MB neurons is TOR-dependent. 

Prior consideration for publication, the manuscript could benefit from some improvements outlined below:

1. Although for the Ca2+ imaging experiments n numbers of brains are stated, this is not the case for results shown in Figs.1-3. The authors should  specify representative of how many examined brains the images shown are.

2. Line 176: the authors should specify which Flybase prediction specifically they are referring to (e.g. Flyatlas etc.).

3. On the images shown in Fig.1c the co-localization with neurons is not as obvious as for glial cells. The authors should include some quantification of co-localization (this could be eventually done using ImageJ), or maybe present a magnified area that better illustrates the co-localization. The authors could also specify which area of the adult brain was imaged in Fig. 1c.

4. Representative images supporting results shown in Fig.S1 could be added to illustrate the dose- and age-dependency of MB neuron activation in response to L-Glu.

5. The authors could comment on why in Fig. 4 in some instances the Ca2+ signal is localized to only one lobe of the MB vs both lobes such as in Fig. 4a? Does this reflect inconsistencies between application of the different solutions or just an image capturing issue?

6. The authors could extend the discussion and incorporate some points about the possible role of MND in glia cells. Since they show that MND is expressed in glia too, they could discuss the significance of this finding.

Reviewer 3 Report

Comments and Suggestions for Authors

The authors have presented an intriguing study demonstrating the role of the amino acid transporter minidiscs (mnd) in mediating Ca2+ homeostasis in mushroom body neurons through the rapamycin (TOR) pathway via L-amino acid sensing. While the findings are interesting, there are points that could be further clarified for publication. Below are my comments:

In Figure 1b, the mnd-common isoform should theoretically exhibit the highest RNA abundance compared to the other three isoforms. However, the RT-PCR results indicate the lowest RNA abundance in mnd-common compared to the other two isoforms. Has the author confirmed these PCR products by sequencing the DNA fragments?

Based on Figure 1c, it appears that mnds are strongly expressed in glial cells. Has the author investigated whether activation of glial cells by leucine sensing is also dependent on mnd expression? These findings would elucidate whether the neuronal activation function of mnd is specific to certain cell types or is more general.

In a previous study (Manière et al., Cell Reports 2016), it was demonstrated that mnds mediate leucine sensing via the GDH pathway in Insulin-Producing Cells. It's intriguing to note that in MBs, the TOR pathway mediates leucine sensing through mnds. Does the author have any insights into how different cells respond to the same stimulation via the same mediator but in distinct ways?

The author should show the efficiency of mnd knockdown by mnddsRNAkk.

Round 2

Reviewer 1 Report

Comments and Suggestions for Authors

The authors have addressed the reviewers’ concerns.

Author Response

- Reviewer 2, pt. 5: The authors have provided an explanation for the inconsistencies in image capture, and included an image in the rebuttal to support their assertion.  They should address this in the Results section where they refer to Fig. 4, as a limitation of their methodology.  It is not necessary to include the added image in the Supplementary Material, but it would be acceptable to do so, if desired.

Response: We are very pleased to include the requested clarification to our manuscript. We have added a sentence in the legends of Fig. 4 (line 353) and S2 (line 469) to clarify why the images may show only one lobe (artifact of the software used to construct the rainbow images).

- Reviewer 3, "Based on Figure 1c.." comments:  All reviewers were rightfully interested in the glial expression mnd.  In their rebuttal to this reviewer, who inquired "whether the activation of glial cells by leucine was dependent on mnd expression?", the authors provided a graph that clearly shoed leucine did not alter the calcium response in glial cells.  This graph should be added as a supplementary figure and a brief mention of this result should be included in the Discussion.

 Response: We are very pleased to include the requested additions to our manuscript. We have added a sentence (line 428) and a new figure S3 (line 472) to emphasize the non-responsiveness of glial cells to leucine.

Thank you very much for these constructive reviews to improve our article.

Best regards,

Dr. Yael Grosjean